# Human Papillomavirus Prevalence in the Prostate and Seminal Vesicles: Does This Virus Have an Etiological Role in the Development of Prostate Cancer?

**DOI:** 10.3390/v17101304

**Published:** 2025-09-26

**Authors:** Rei Shinzawa, Kazuyoshi Shigehara, Hiroki Nakata, Shingo Takada, Kotaro Fukukawa, Yuki Kato, Tomomi Nakagawa, Shohei Kawaguchi, Kouji Izumi, Atsushi Mizokami

**Affiliations:** 1Integrative Cancer Therapy and Urology, Division of Cancer Medicine, Graduate School of Medical Science, Kanazawa University, Kanazawa 920-0934, Japan; bunchof@gmail.com (R.S.);; 2Department of Urology, Kanazawa Medical Center, Kanazawa 920-8650, Japan

**Keywords:** papillomavirus, prostate cancer, seminal vesicle, p16, E6/E7 mRNA

## Abstract

Human papillomavirus (HPV) is common in both sexes and is also detected in male urine and semen. However, its exact origin and its etiological role in the male genital tract remain unclear. A total of 157 formalin-fixed paraffin-embedded tissues from 156 primary prostate cancer lesions and one metastatic lesion were analyzed. HPV-DNA was detected using a nested PCR, and HPV genotyping was performed using flow-through hybridization for positive cases. In situ hybridization (ISH) was used to localize HPV-DNA, whereas HPV-E6/E7 mRNA ISH and p16^INK4a^ immunohistochemistry were conducted on high-risk (HR) HPV-positive samples. A nested PCR analysis demonstrated that HPV-DNA was detected in 9.6% of prostate cancers and 0% of seminal vesicles. HR-HPV was observed in 4.5% of the samples. Unknown type was the most common genotype. Of the genotypes which could be identified in the genotyping assay, HPV44 was the most prevalent. HPV prevalence was significantly higher in patients with high-grade groups. Among 15 HPV-positive cases, HPV-DNA was found in 9 cancerous and 10 non-cancerous lesions. E6/E7 mRNA was expressed in 6 of 7 HR-HPV-positive cases, while p16^INK4a^ expression was weak or absent in all cases. HPV can infect prostate tissue and may contribute to carcinogenesis in some cases, but p16^INK4a^ was not a consistent surrogate for E7 expression.

## 1. Introduction

Human papillomavirus (HPV) infection is a common sexually transmitted infection. It is equally prevalent in men and women [1]. In men, the most frequent infection sites are the external genitalia, such as the glans penis, penile shaft, coronal sulcus, and inner foreskin; however, it also occurs in the urinary tract, urine, and semen [2,3,4,5]. Recently, we demonstrated that HPV prevalence in semen among Japanese infertile men was 12.5% [5]; however, it remains unclear whether the origins of HPV detected in seminal samples, are the urethra or genital tracts, such as the prostate and seminal vesicle.

Although HPV is a known causative agent of uterine cervical cancer, several studies indicate that HPV infection has a potential role in the development of other cancers, including skin, oropharyngeal, and penile cancers [6,7,8]. It has been associated with approximately 10% of all cancer cases [9]. Although the association between HPV infection and prostate cancer remains inconclusive, a meta-analysis involving 3122 cases reported an HPV detection rate of 25.8% in prostate cancer patients, 17.4% in those with benign prostatic hyperplasia (BPH), and 9.2% in healthy controls [10]. Thus, HPV is detectable in prostate tissue, and HPV prevalence is likely significantly higher in prostate cancer patients compared with BPH and controls. Nonetheless, the etiological role of HPV infection in the development of prostate cancer remains unclear.

In this study, we examined the HPV-DNA prevalence and genotypes in prostate cancer and seminal vesicle samples among 157 Japanese patients with prostate cancer. The relationship between HPV infection and the clinical characteristics of prostate cancer was assessed. Moreover, for HPV-positive samples, we performed HPV-DNA in situ hybridization (ISH) to observe the localization of HPV-DNA.

HPV is a nonenveloped icosahedral virus containing a double-stranded circular DNA genome. When a part of the HPV-DNA is integrated into the host genome, HPV-E6 and E7 proteins are expressed, which is determined to be central to the carcinogenic process. The E6 protein inhibits the tumor suppressor p53, thereby suppressing apoptosis. The E7 protein binds to and activates calpain, which cleaves retinoblastoma (Rb) protein and promotes its proteasomal degradation [11,12]. As an additional analysis, we performed HPV-E6/E7 mRNA ISH analysis in all high-risk (HR) HPV-positive samples to assess an etiological role in the development of prostate cancer, as transcription of E6 and E7 generally occurs together.

Under normal conditions, the cyclin-dependent kinase inhibitor p16^INK4a^ activates Rb and halts cell proliferation. In HPV-associated tumors, Rb protein is inactivated through degradation by calpain. Calpain is activated by the E7 protein, which is expressed after HPV-DNA is integrated into the host genome. This results in a loss of negative feedback and overexpression of p16^INK4a^, which persists even during cell proliferation. Based on this mechanism, p16^INK4a^ has been widely recognized as a surrogate marker for HPV-related malignancies [13]. We also evaluated p16^INK4a^ expression in all HR-HPV-positive samples by immunohistochemistry (IHC) to determine whether it is a surrogate marker for E7 expression in prostate cancer with HR-HPV infection.

## 2. Materials and Methods

### 2.1. Subjects

Japanese prostate cancer patients who underwent robot-assisted radical prostatectomy or metastasectomy for prostate cancer at the Department of Urology, Kanazawa University Hospital, Kanazawa, Japan, between July 2017 and May 2019 were enrolled. The study was conducted following a protocol approved by the Ethics Committee of the Graduate School of Medical Sciences, Kanazawa University (Approval No. 2019-088). Information regarding the study was disclosed, and clinical data, such as patient background and tumor characteristics, were collected from the available medical records. Formalin-fixed paraffin-embedded (FFPE) prostate cancer samples were obtained for all cases; however, only one case had a metastatic lesion available, as the primary tumor could not be obtained. In addition, FFPE samples of the seminal vesicle were also collected. In total, 157 prostate cancer specimens (156 primary prostate cancer lesions and one metastatic lesion) and 156 seminal vesicle specimens were analyzed.

### 2.2. HPV-DNA Test and Genotyping

FFPE blocks prepared with the prostate cancer and seminal vesicle specimens were sectioned at a thickness of 5 μm. After deparaffinizing, DNA was extracted from each sample using the Pinpoint Slide DNA Isolation System (Zymo Research, Orange, CA, USA). The quality of the DNA was assessed by confirming the amplification of the β-globin gene by the polymerase chain reaction (PCR). HPV-DNA was detected using nested PCR targeting the L1 gene, with MY09/MY11 as the outer primers and GP5+/GP6+ as the inner primers [14]. For HPV-positive samples, HPV genotyping was performed using the 21-HPV GenoArray Diagnostic Kit (HybriBio, Hong Kong, China) [15], which uses the flow-through hybridization technique to identify a total of 21 HPV genotypes, including 15 high-risk HPV (HR-HPV) (16, 18, 31, 33, 35, 39, 45, 51, 52, 53, 56, 58, 59, 66, and 68) and 6 low-risk HPV (LR HPV) (6, 11, 42, 43, 44, and CP8304) types. Samples that tested positive by nested PCR, but yielded no detectable genotype using the GenoArray kit, were classified as having an unknown type.

### 2.3. HPV-DNA In Situ Hybridization

ISH analysis was used to confirm the localization of HPV-DNA in all of the HPV-positive samples. HPV-positive FFPE blocks were sectioned at 5-μm thickness, and hybridized with an HPV-DNA probe (Y1404; Dako, Carpinteria, CA, USA) based on the manufacturer’s protocol for the Dako GenPoint™ System [8]. This ISH analysis detects 13 HR-HPV genotypes (HPV types 16, 18, 31, 33, 35, 39, 45, 51, 52, 56, 58, 59, and 68). As a negative control, some of the HPV-negative samples were processed simultaneously. The staining distribution of the ISH signals was scored as negative (−; no staining), positive (+; visible only at high magnification), and strongly positive (2+; clearly visible at low magnification).

### 2.4. HPV-E6/E7 mRNA In Situ Hybridization

HPV-E6/E7 mRNA ISH analysis was conducted to assess an etiological role for the development of prostate cancer in all HR-HPV-positive samples. E6/E7 mRNA expression was investigated using RNAscope^®^ technology (Advanced Cell Diagnostics, Newark, CA, USA). This procedure was conducted with a specific probe (Code: 312591; Advanced Cell Diagnostics), targeting 18 HR-HPV genotypes (HPV types 16, 18, 26, 31, 33, 35, 39, 45, 51, 52, 53, 56, 58, 59, 66, 68, 73, and 82). As a negative control, E6/E7 mRNA expression was investigated in some of the HPV-negative samples. The staining distribution of the ISH signals was evaluated as described above; negative (−; no staining), positive (+; visible only at high magnification), and strongly positive (2+; visible clearly at low magnification).

### 2.5. Immunohistochemistry of p16^INK4a^, a Surrogate Marker of the E7 Protein

HPV-positive FFPE blocks were sectioned at 5-μm thickness, and antigen retrieval was performed by incubating the samples in 20 mM Tris-HCl (pH 9.0) at 95 °C for 20 min, followed by cooling at room temperature for 20 min. The samples were incubated overnight with a mouse monoclonal p16^INK4a^ antibody (ACR3231A; BIOCARE Medical, Pacheco, CA, USA) diluted 1:100. Next, the specimens were incubated with a peroxidase-labeled secondary antibody for 30 min. The expression of p16^INK4a^ was observed following DAB staining (SK-4103; Vector LABORATORIES, Newark, CA, USA), and the slides were counterstained with hematoxylin. Some HPV-negative samples and HPV16-positive penile cancer samples [8] were available for IHC of p16^INK4a^ as negative and positive controls, respectively. We assessed the IHC staining as follows: negative (−; no staining), positive (+; visible only at high magnification), and strongly positive (2+; clearly visible at low magnification).

### 2.6. Statistical Analysis

Patient background characteristics were compared using the Mann–Whitney U test or the Chi-squared test, as appropriate. Comparisons between the Grade Group (GG) and HPV-positive and -negative cases were performed using the Chi-squared test. All statistical analyses were performed using IBM SPSS Statistics version 29 (IBM Corp., Armonk, NY, USA), and a *p*-value of <0.05 was considered statistically significant.

## 3. Results

### 3.1. Patient Characteristics and HPV Prevalence

The median age and serum prostate-specific antigen (PSA) levels of the 157 patients with prostate cancer were 68 years (range, 48–76 years) and 7.05 ng/mL (range, 2.42–74.2 ng/mL), respectively (Table 1). Nineteen (12.1%) cases were histologically GG1, 118 (75.2%) were GG2–3, and 20 (12.7%) were GG4–5 cancer. Many of the patients (87.9%) had localized prostate cancer (<Tumor category 2). None of the patients had urethritis, syphilis, or human immunodeficiency virus infection at the time of specimen collection.

HPV-DNA was detected in 15 cases (9.6%) of prostate cancer and 0 (0%) of the seminal vesicle samples (Table 2). HR-HPV was prevalent in 7 prostate cancer cases (4.5%). An unknown type, which was not determined by the flow-through hybridization of 21 types, was the most common genotype (6 cases). Of the genotypes which could be identified in the genotyping assay, HPV44 was the most frequent, followed by HPV31, 52, and 58. HPV52 was detected in one metastatic lesion.

When patient backgrounds were stratified by HPV-DNA status, HPV prevalence in the patients with high GG (4–5) was significantly higher compared with those exhibiting low to intermediate GG (1–3) (*p* = 0.0214) (Table 1).

### 3.2. HPV-DNA In Situ Hybridization

In the 15 HPV-positive samples, HPV-DNA signals were observed in cancerous lesions in 9 (60%) cases. Positive signals were obtained in 2 cases, and 7 cases were strongly positive (Figure 1, Table 3). Strong ISH signals (strongly positive) were observed in 4 HR-HPV-positive cases, whereas weak signals (positive) were evident in one case. Moreover, ISH signals were also present in the cells of normal glandular tissue in non-cancerous lesions in 10 cases (Figure 1). HPV-DNA signals were even observed in samples that were positive for HPV genotypes (type 44, 66, and UK) not covered by the ISH probe. As a negative control, HPV-DNA signals were undetectable in the HPV-negative samples.

### 3.3. HPV-E6/E7 mRNA Expression

We performed HPV-E6/E7 mRNA ISH analysis to determine the etiological role for the development of prostate cancer in HR-HPV-positive samples. Strong signals were present in the nuclei of tumor cells among 6 of 7 HR-HPV-positive samples (Table 4, Figure 2). Among HR-HPV-positive samples with HPV-DNA ISH signals in cancerous lesions, 4 exhibited strong expression of E6/E7 mRNA. Furthermore, ISH signals of E6/E7 mRNA were also detected in two cases without HPV-DNA ISH signals. On the other hand, E6/E7 mRNA was not observed in one HR-HPV-positive sample and the HPV-negative samples.

### 3.4. p16^INK4a^ Protein Expression

p16^INK4A^ expression was detected in 4 HR-HPV-positive samples (Table 4); however, p16^INK4a^ overexpression was not observed, and it was weak in all cases (Figure 3). The remaining 3 cases showed no p16^INK4a^ expression. On the other hand, p16^INK4a^ was not expressed in the HPV-negative samples.

## 4. Discussion

We found that the prevalence of HPV-DNA was 9.6% (15/157 cases) among prostate cancer patients based on a nested PCR analysis. Numerous studies have demonstrated a wide range of HPV prevalence in prostate cancer patients, from 0% to 75% worldwide [10,16,17,18,19,20], and from 0% to 16% in Japan [18,19]. This variability may be the result of small cohorts, differences in HPV detection methods, and differences in the surveyed regions. The prevalence of HPV varies by region, particularly with East Asian populations, which generally show lower rates compared with the global average [20]. Although HPV prevalence in the present study was not considered high, it was consistent with the previous findings in Japan [18,19]. Moreover, the prostate was one of the HPV infectious sites in men; however, HPV-DNA was not detected in the seminal vesicle. To our knowledge, there are no previous reports examining HPV infection in the seminal vesicles. Currently, there is no evidence indicating that HPV can infect the seminal vesicles.

In the present study, the most frequent HPV genotype was the UK type, followed by type 44 (5 cases), and types 31, 52, and 58 (2 cases, respectively) (Table 2). A meta-analysis in 2019 involving 5546 prostate cancer patients indicated that HPV16 was the most prevalent, followed by types 31 and 58 [21]. Therefore, the HPV type distribution was different from our results. Many studies included in this meta-analysis only evaluated HR-HPV types; thus, LR genotypes, such as HPV 44, may have been categorized as “unknown types.” Indeed, a previous large population study including 803 Japanese patients evaluating HPV prevalence in urine samples demonstrated that HPV types 70, 71, 84, and 90, which are not covered by the HPV genotyping method used in the present study, were frequently detected [22]. Although the HPV type distribution between the urine and prostate specimens is not always consistent, the HPV genotypes detected in the prostate samples may also vary by region or country. Our results suggest that LR-HPV infection, including UK types, may also be relatively common in the prostate. On the other hand, HPV16, which is the most common oncogenic type observed in many HPV-associated cancers, was not detected in the present study. This discrepancy may also be due to differences in target subjects. However, the number of patients was small for an epidemiological study; thus, further studies including a large number of subjects are needed to clarify HPV prevalence in the prostate.

Prostate cancer is histologically classified into GG1–5. According to the National Comprehensive Cancer Network guidelines, GG1 corresponds to a low or very low risk, GG2–3 is at intermediate risk, and GG4–5 represents high or very high risk. In our study, patients were grouped into GG1, GG2–3, and GG4–5 categories. The HPV detection rate was higher in cases with high GG (Table 1). A previous study including 95 prostate cancer cases revealed that a higher HPV prevalence was associated with cases exhibiting a high Gleason score [23]. Other studies have also demonstrated an association between HPV infection and high Gleason score in prostate cancer [24,25]. HPV infection may also be correlated with tumor aggressiveness.

In the present study, we performed ISH analysis to confirm the localization of HPV-DNA in prostate tissue. Among the 15 HPV-positive samples, HPV-DNA signals were detected in the cancerous lesions of 9 cases, and in non-cancerous lesions of 10 cases (Table 3). Unexpectedly, HPV infection was also found in non-cancerous lesions in the prostate cancer samples. Two HR-HPV-positive cases showed no signals in cancerous lesions, but in non-cancerous lesions. Overall, ISH signals were observed in 12 (80%) of 15 HPV-positive cases based on PCR analysis, and the signals were observed in all HR-HPV-positive specimens. These results confirm the presence of HPV-DNA in prostate tissue. On the other hand, HPV-DNA signals were detected even in some samples that were positive for HPV genotypes which were not covered by the ISH probe. This may be due to cross-hybridization with the 44, 66, and unknown types, since HPV-DNA signals were not observed in HPV-negative samples. The similar cross-reactions of HPV-DNA in the ISH procedure used in this study have been described previously [26].

In HPV-associated malignancies, such as penile cancer, oropharyngeal cancer, and anal cancer, persistent infection with HR-HPV results in the integration of HPV-DNA into the host genome, the expression of the E6 and E7 oncogenes, and an increased risk of carcinogenesis [27,28,29]. The association between HPV and cancer and the expression of the E6 and E7 genes has been evaluated to determine their etiological role in carcinogenesis [26,27,28]. In the present study, we found strong E6/E7 mRNA expression in 6 of 7 HR-HPV-positive samples, supporting a carcinogenic role for HR-HPV infection (Table 4). In the remaining case (case 7, HPV58), HPV-DNA was detected in the cancerous lesions; however, E6/E7 mRNA was not expressed. This suggests that HPV58 may have been an episomal infection in this case. Only one study demonstrated that the E7 protein was expressed in all HR-HPV-positive samples by IHC [17]. Moreover, a recent review proposed the potential involvement of the E6 and E7 proteins in the oncogenic mechanism of HPV infection [30]; however, studies examining E6/E7 expression in prostate cancer are currently limited. Therefore, further studies are needed to establish the oncogenic role of the E6/E7 protein in prostate cancer.

p16^INK4a^ has been widely accepted as a surrogate marker of E7 expression, and this protein is markedly expressed in many HPV-associated cancers; however, our results indicate that p16^INK4a^ overexpression is not observed in all cases (Table 4). A noteworthy observation was the discordance between E6/E7 mRNA expression and p16^INK4A^ expression. It has been well documented that some oropharyngeal cancers lack p16^INK4A^ expression despite the functional inactivation of Rb by the E7 protein [31,32]. This may result from the difference in viral load and mutations or deletions in the p16^INK4A^ gene (CDKN2A). In particular, CDKN2A is more frequently methylated in prostate cancer tissue compared with normal tissue [33], suggesting that p16^INK4A^ expression may be epigenetically suppressed in malignant lesions. On the other hand, one study reported an HR-HPV prevalence of 19.6% in prostate cancer and observed p16^INK4A^ overexpression in many HPV-positive samples [34], which differs from our results. Few studies have examined the relationship between HPV infection and p16 ^INK4a^ expression in prostate cancer. Although further studies are needed, our results suggest that p16 ^INK4a^ may not always be a surrogate marker for HPV-DNA integration in prostate cancer.

There are some limitations in the present study. Initially, the number of HR-HPV-positive samples was extremely limited, and E6/E7 mRNA and p16 ^INK4a^ expression were only evaluated in 7 cases. In addition, we collected all of samples from the patients with prostate cancer in the present study, and a normal control group was not included. This limitation makes it difficult to establish a baseline prevalence in healthy prostate tissue, and leads to difficulty in distinguishing between the mere presence of the virus and its oncogenic etiological role. Therefore, further studies including non-cancerous tissues are required to solve this limitation. Alternatively, uses of cultured normal human prostate infected with HPV cells as a control may be considered. Furthermore, in addition to E6 and E7 oncoproteins, chronic inflammation is considered to be an important factor in oncogenic mechanisms of prostate cancer [30]. However, there were no data of inflammation and sexually transmitted infections. Therefore, further studies with a large number of subjects are needed to establish an etiological role of HPV infection in prostate cancer. On the other hand, we examined the carcinogenic role of HPV infection in prostate cancer using various molecular analyses, such as PCR, HPV-DNA ISH, E6/E7 mRNA ISH, and IHC, which is a strength.

## 5. Conclusions

HPV-DNA was detected in 9.6% of prostate cancer cases and 0% of the seminal vesicle specimens. ISH analysis revealed that HPV-DNA was present in the cancerous and/or non-cancerous lesions, suggesting that HPV may infect the prostate. In some cases of HR-HPV infection, HPV-E6/E7 mRNA was expressed in the cancer tissue. On the other hand, p16^INK4a^ expression was weak or absent in all cases, and p16^INK4a^ was not a consistent surrogate for E7 expression in prostate cancer. HPV infection occurs in the prostate, and it may play an etiological role in the development of prostate cancer, although this has only been tested in a limited number of cases to date.

## Figures and Tables

**Figure 1 viruses-17-01304-f001:**
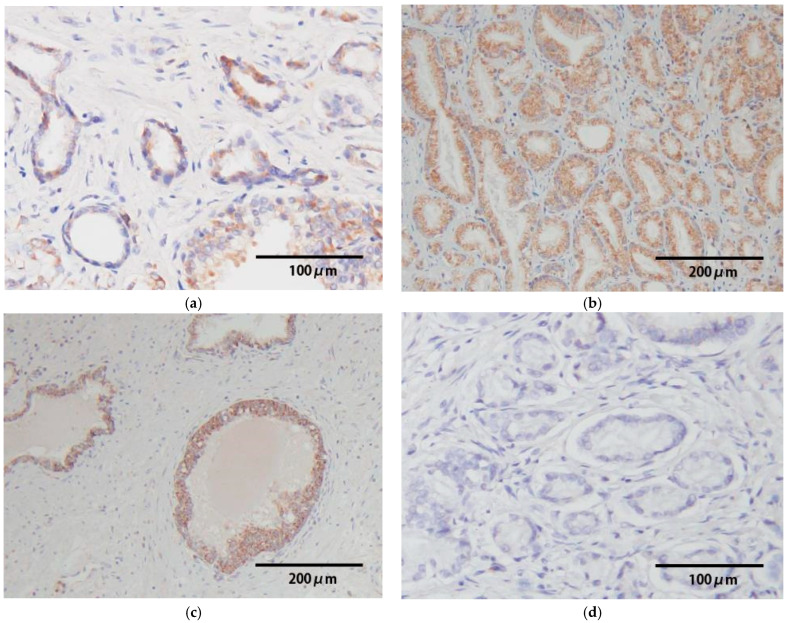
Results of HPV-DNA in situ hybridization (ISH) are indicated. HPV-DNA signals were observed in cancerous lesions in 9 cases. Positive signals ((**a**), ×200; case 14) and strong positive signals ((**b**), ×100; case 8) were shown. ISH signals were also present in the cells of normal glandular tissue in non-cancerous lesions. ((**c**), ×100; case 9). HPV-DNA signals were undetectable in the HPV-negative samples. ((**d**), ×200).

**Figure 2 viruses-17-01304-f002:**
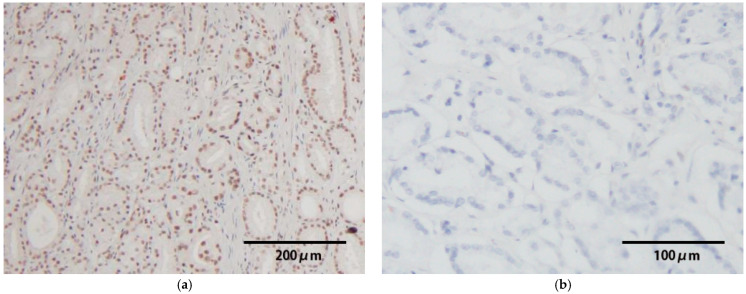
Results of E6/E7 mRNA in situ hybridization are indicated. Strong signals were present in the nuclei of tumor cells among 6 of 7 HR-HPV-positive samples. ((**a**), ×100; case 8) On the other hand, E6/E7 mRNA was not observed in the HPV-negative samples. ((**b**), ×200).

**Figure 3 viruses-17-01304-f003:**
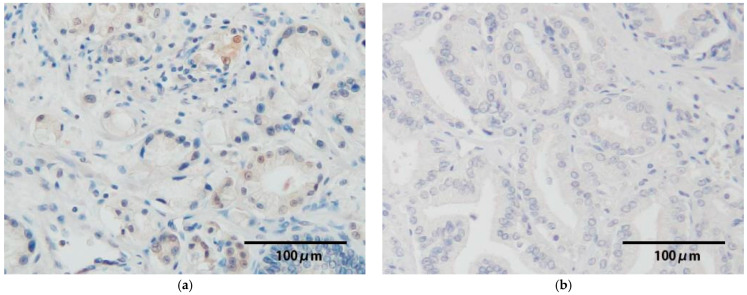
Results of p16^INK4a^ immunohistochemistry are indicated. p16^INK4A^ expression was detected in HR-HPV-positive samples; however, p16INK4a overexpression was not observed, and it was weak ((**a**), ×200; case 8). On the other hand, p16^INK4a^ was not expressed in the HPV-negative samples ((**b**), ×200).

**Table 1 viruses-17-01304-t001:** Patient backgrounds and tumor characteristics.

Characteristics	n = 157	HPV(+) (n = 15)	HPV(−) (n = 142)	*p*
Age (median, range)	68 (48–76)	70 (48–76)	65 (56–76)	0.981
PSA, ng/mL (median, range)	7.05 (2.42–74.2)	6.24 (3.00–16.54)	7.06 (2.42–74.2)	0.375
Grade Group (n, %)				
1	19 (12.1%)	0 (0%)	19 (13.4%)	
2–3	118 (75.2%)	10 (66.7%)	108 (76.1%)	
4–5	20 (12.7%)	5 (33.3%)	15 (10.6%)	0.0214
T category (n, %)				
pT0	2 (1.3%)	0	2 (1.4%)	
pT2	136 (86.6%)	12 (80%)	124 (87.3%)	0.455
pT3	15 (9.6%)	3 (20%)	12 (8.5%)	
pT4	4 (2.5%)	0	4 (2.8%)	

HPV, human papillomavirus; PSA, prostate-specific antigen; T, tumor.

**Table 2 viruses-17-01304-t002:** HPV-DNA prevalence and genotyping in prostate cancer specimens and seminal vesicle samples.

PCR Results	Prostate Cancer	Seminal Vesicle
Any HPV	15 (9.6%)	0 (0%)
HR-HPV	7 (4.5%)	0 (0%)
The HPV genotype	n	
31	2	
44	5	
52	2	
58	2	
66	1	
Unknown ^1^	6	

PCR, polymerase chain reaction; HPV, human papillomavirus; HR, high-risk. ^1^ Specimens that tested positive by nested PCR, but yielded no detectable genotype by the GenoArray kit, were classified as having an unknown type.

**Table 3 viruses-17-01304-t003:** Summary of the HPV genotypes and HPV-DNA in situ hybridization in HPV-positive specimens.

No	Age	GG	T	Genotype	HPV Risk	HPV-DNA ISH	
						Normal Lesion	Cancerous Lesion
1	48	3	2c	31	HR	2+	−
2	63	2	3a	31	HR	2+	2+
3	74	5	3b	44	LR	+	2+
4	59	2	2c	44	LR	2+	2+
5 *	59	5	3b	52	HR	−	+
6	71	2	2c	UK	Unknown	+	−
7	72	4	2a	58	HR	2+	2+
8	76	2	2c	66	HR	−	2+
9	72	3	2b	52/44	HR/LR	2+	2+
10	63	4	2a	58/44	HR/LR	+	−
11	64	4	2c	44	LR	−	−
12	70	3	2c	UK	Unknown	2+	2+
13	72	2	2c	UK	Unknown	−	−
14	72	2	2b	UK	Unknown	+	+
15	65	2	2c	UK	Unknown	−	−

HPV, human papillomavirus; GG, Grade Group; T, tumor; HR, high-risk; LR, low-risk; UK, unknown. ISH, in situ hybridization. Staining score; negative (−; no staining), positive (+; visible only at high magnification), and strongly positive. (2+; visible clearly at low magnification). * A case in which HPV-DNA was detected in metastatic prostate cancer lesion.

**Table 4 viruses-17-01304-t004:** Summary of HPV genotypes and HPV-E6/E7 expression in HR-HPV-positive specimens.

No	Genotype	HPV Risk	E6/E7 Oncogenic Protein	
			E6/E7 mRNA ISH	p16^INK4a^ Protein
1	31	HR	2+	+
2	31	HR	2+	−
5 *	52	HR	2+	−
7	58	HR	−	+
8	66	HR	2+	+
9	52/44	HR/LR	2+	+
10	58/44	HR/LR	2+	−

HPV, human papillomavirus; HR, high-risk; LR, low-risk; RNA, ribo nucleic acid. ISH, in situ hybridization. Staining score; negative (−; no staining), positive (+; visible only at high magnification), and strongly positive (2+; visible clearly at low magnification). * A case in which HPV-DNA was detected in metastatic prostate cancer lesion.

## Data Availability

The data that support the findings of this study are available on request from the corresponding author. The data are not publicly available due to privacy or ethical restrictions.

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
