# Peer review of "Human Papillomavirus Prevalence in the Prostate and Seminal Vesicles: Does This Virus Have an Etiological Role in the Development of Prostate Cancer?"

_viruses, 2025, doi:10.3390/v17101304_

Round 1
Reviewer 1 Report
Comments and Suggestions for Authors
This paper is an additional potentially intriguing study about the association between HPV infections and prostate cancer.
In its present form, this study gives us a piece of information about territorial, regional presence of HPVs. However, it is very curious that these authors were not able to detect HPV16 believed the strongest and most significant positive association with prostate cancer. Is there some explanation? I suggest these authors to use specific primers for HPV16 and HPV18.
Of note, the large burden of disease justifies the need to identify modifiable risk factors. About 20% of all adult cancers results from chronic inflammation. This relationship have led researchers to investigate the role of sexually transmitted infections (STIs) in the development of prostate cancer.
May the authors provide some information about the most common sexually transmitted infections such as Treponema pallidum, Neisseria gonorrhoeae, Clamydia tracomatis, Trichomonas vaginalis, human immunodeficiency virus (HIV) infections in their cases?
Line 93: the authors wrote: The viral genome encodes several genes, including E1, E2, E3, E4, E5, E6,…. E3??? Which clade??????
Furthermore, I presume that the authors wanted to write ….encodes several oncoproteins?????
Lines 96-97: the authors wrote: … E7 protein promotes cellular proliferation by inactivating the retinoblastoma protein [14]. Although E7-mediated mechanism is still to be elucidated in deep, it is believed that E7 is not responsible for pRb inactivation but E7 promotes the accelerated proteasomal degradation of pRb via calpain system. I suggest the authors to describe better this mechanism by citing Dornell et al’s (doi: 10.1074/jbc.M706860200) and De Falco et al.’s (doi: 10.1080/01652176.2024.2387072).
In conclusion, I suggest the authors to strengthen this study with deeper molecular investigations such as reverse transcription PCR for detection of the oncoproteins (of course, if possible).
Author Response
Comments 1: This paper is an additional potentially intriguing study about the association between HPV infections and prostate cancer.
In its present form, this study gives us a piece of information about territorial, regional presence of HPVs. However, it is very curious that these authors were not able to detect HPV16 believed the strongest and most significant positive association with prostate cancer. Is there some explanation? I suggest these authors to use specific primers for HPV16 and HPV18
Response 1: Thank you for your kind comments. As a reviewer says, HPV16 is the most common in carcinogenesis of many HPV infection associated carcinoma. However, HPV16 was not detected in the present study. In this study, we used an HPV detection kit (HPV GenoArray Diagnostic Kit), which is capable of detecting multiple genotypes including HPV16 and HPV18. Although not shown in the manuscript, we performed uniplex PCR using specific probes for HPV16 and HPV18, and however both were negative. In discussion, we explained this discrepancy to “Our results suggest that LR HPV infection, including UK types, may also be relatively common in the prostate. On the other hand, HPV16, which is the most common oncogenic type observed in many HPV-associated cancers, was not detected in the present study. This discrepancy may be also due to difference of target subjects. However, the number of patients was small for an epidemiological study; thus, further studies including a large number of subjects are needed to clarify HPV prevalence in the prostate.”.
Comments 2: Of note, the large burden of disease justifies the need to identify modifiable risk factors. About 20% of all adult cancers results from chronic inflammation. This relationship have led researchers to investigate the role of sexually transmitted infections (STIs) in the development of prostate cancer.
Response 2: Thank you for your comments. As the reviewer pointed out, many reports have also described a positive association between prostatitis and prostate cancer, and then its relationship may be important when examining the carcinogenic mechanism of prostate cancer. Unfortunately, data on chronic inflammation and STIs were not included in the present study. Therefore, in the present study, we could not mention involvement of inflammation in carcinogenesis of prostate cancer. In the limitations of discussion section, we added the comments as following; “Furthermore, the absence of a normal control group is not included. In addition to E6 and E7 oncoproteins, chronic inflammation is considered to be an important factor in oncogenic mechanisms [35]. However, there were no data of inflammation and sexually transmitted infections. Therefore, further studies with a large number of subjects are needed to establish an etiological role of HPV infection in prostate cancer.”
Comments 3: May the authors provide some information about the most common sexually transmitted infections such as Treponema pallidum, Neisseria gonorrhoeae, Clamydia tracomatis, Trichomonas vaginalis, human immunodeficiency virus (HIV) infections in their cases?
Response 3: Thank you for your comments. As a reviewer says, there are some reports to indicate relationship between sexually transmitted infections (STIs) and prostate cancer. In the present study, information regarding past histories of STIs among the patients is absent. At least, none had HIV infection, syphilis, or urethritis in the present study. Therefore, a sentence of “None of the patients had urethritis, syphilis, or human immunodeficiency virus infection at the time of specimen collection.” was added in the results.
Comments 4: Line 93: the authors wrote: The viral genome encodes several genes, including E1, E2, E3, E4, E5, E6,…. E3??? Which clade??????
Furthermore, I presume that the authors wanted to write ….encodes several oncoproteins?????
Response 4: Thank you for your comments. These sentences were revised in the introduction section.
Comments 5: Lines 96-97: the authors wrote: … E7 protein promotes cellular proliferation by inactivating the retinoblastoma protein [14]. Although E7-mediated mechanism is still to be elucidated in deep, it is believed that E7 is not responsible for pRb inactivation but E7 promotes the accelerated proteasomal degradation of pRb via calpain system. I suggest the authors to describe better this mechanism by citing Dornell et al’s (doi: 10.1074/jbc.M706860200) and De Falco et al.’s (doi: 10.1080/01652176.2024.2387072).
Response 5: Thank you for your comments. Based on the literature you provided, we revised descriptions regarding the functions of E6/E7 protein in the introduction section.
Comments 6: In conclusion, I suggest the authors to strengthen this study with deeper molecular investigations such as reverse transcription PCR for detection of the oncoproteins (of course, if possible).
Response 6: Thank you for your comments. I agree that performing reverse transcription PCR for oncoprotein detection could provide additional insights. However, we examined the carcinogenic role of HPV infection in prostate cancer using various molecular analyses, such as PCR, HPV-DNA ISH, E6/E7 mRNA ISH, and IHC, which is acceptable for the current assessment. Nevertheless, I acknowledge that further molecular investigations could be considered in future studies. These points have been described in discussion.
Reviewer 2 Report
Comments and Suggestions for Authors
This manuscript by Shinzawa et al examines the presence of HPV in cancer tissues of patients with prostate cancer. While this topic has been examined to some degree by others, there is no strong evidence to suggest the connection between HPV infection as an etiologic factor in development of prostate cancer. The authors examined archived tissue samples. While the authors are asking an important question, I am afraid the data is not convincing with regards to the potential role of HPV infection in prostate cancer. Here are my comments:
- The biggest problem in this manuscript is that there are no normal control samples from healthy individuals. I understand that this is difficult given that normal prostate tissues may be hard to come by unless tissues were available from some kind of a surgical procedure not related to cancer. The authors genotype the HPV types in the prostate cancer samples and found a bunch of different types represented. However, these could be contamination from the skin as different HPV types are readily swabbed and typed as seen in other studies. I am not convinced that the HPV typing is specific in relation to the cancer being studied.
- The authors mention finding HPV44 in the genotyping. However, HPV44 is a low-risk type not related to cancer causation. Does the genotyping array detect HPV44 ? I did not see that in the materials and methods.
- The HPV in situ hybridization, E6/E7 mRNA in situ hybridization, and p16 staining are not convincing. Staining alone is rather subjective when visualized as not all areas stain uniformly. The staining could be quantified and significance calculated. However, there was no measures taken to quantify the staining. The staining data as it stands is not convincing. It would be good to have some control tissues or even cultured cells dervied from normal prostate as comparison to determine the specificity of staining with regards to HPV.
- Were the seminal vesicle samples derived from the patients the same time the cancer tissues were derived ? It is not clear from the writing at which stage those samples were generated.
- Most p values calculated and presented within the manuscript are not significant. Therefore the biology pertaining to the question being asked is absent.
Author Response
Comments 1: The biggest problem in this manuscript is that there are no normal control samples from healthy individuals. I understand that this is difficult given that normal prostate tissues may be hard to come by unless tissues were available from some kind of a surgical procedure not related to cancer. The authors genotype the HPV types in the prostate cancer samples and found a bunch of different types represented. However, these could be contamination from the skin as different HPV types are readily swabbed and typed as seen in other studies. I am not convinced that the HPV typing is specific in relation to the cancer being studied.
Response 1: Thank you for your comments. As the reviewer pointed out, it would be desirable to compare with a control group; however, normal prostate tissue is not available (Prostatectomy can’t be performed for the patients without prostate cancer). We added this point as a limitation. In the present study, we examined the carcinogenic role of HPV infection in prostate cancer using various molecular analyses, such as PCR, HPV-DNA ISH, E6/E7 mRNA ISH, and IHC. In particular, ISH and IHC analyses are largely free of contamination. Therefore, we believe that our data is likely to be fully acceptable, although further studies are needed.
Comments 2: The authors mention finding HPV44 in the genotyping. However, HPV44 is a low-risk type not related to cancer causation. Does the genotyping array detect HPV44 ? I did not see that in the materials and methods.
Response 2: Thank you for your comments. As the reviewer mentions, HPV44 is a low-risk type. Our description may have misleadingly implied that HPV44 is a high-risk HPV type. To avoid any confusion, we corrected the statements in abstract and results.
Comments 3: The HPV in situ hybridization, E6/E7 mRNA in situ hybridization, and p16 staining are not convincing. Staining alone is rather subjective when visualized as not all areas stain uniformly. The staining could be quantified and significance calculated. However, there was no measures taken to quantify the staining. The staining data as it stands is not convincing. It would be good to have some control tissues or even cultured cells dervied from normal prostate as comparison to determine the specificity of staining with regards to HPV.
Response 3: Thank you for your comments. I agree with the reviewer that quantification of the staining and comparison with control tissues would strengthen the analysis. Unfortunately, as mentioned in my response to Comment 1, normal prostate tissue was not available, so such experiments could not be conducted in this study. The absence of a control group was noted as a limitation. Although IHC and ISH analyses have the disadvantage of not being able to quantify the results, these methods are widely used in many previous studies.
Comments 4: Were the seminal vesicle samples derived from the patients the same time the cancer tissues were derived ? It is not clear from the writing at which stage those samples were generated.
Response 4: Thank you for your comments. In robot-assisted radical prostatectomy for the patients with prostate cancer, the prostate and seminal vesicles are always removed simultaneously.
Comments 5: Most p values calculated and presented within the manuscript are not significant. Therefore the biology pertaining to the question being asked is absent.
Response 5: Thank you for your comments. In this study, the only statistically significant correlation observed was between the Gleason group and the presence of HPV-DNA. Including low-risk HPV types, there were 15 cases, and when considering high-risk HPV types (including cases co-infected with low-risk types), there were 7 cases. The limited number of cases is considered a limitation of this study. This is likely influenced by the relatively low HPV positivity rate in Eastern Asia, including Japan, compared to other regions. These limitations are described in discussion.
Reviewer 3 Report
Comments and Suggestions for Authors
The manuscript is of great interest but it is written carelessly. The inaccuracies and errors should be corrected.

Comments on the Quality of English Language
It should be accepted with minor corrections
Author Response
Comments 1: Section 2.3. Line 83. No Sentence Start.
Response 1: Thank you for your comments. I revised it to ‘ISH analysis was…’.
Comments 2: Section 2.4. Lines 92-97 contain generally known information that does not pertain to the Materials and Methods section. Therefore, it is advisable to move lines 92-97 to the Introduction section.
Response 2: In accordance with reviewer’s suggestion, this sentence was moved to the Introduction section.
Comments 3: It is advisable to indicate why E6/E7 mRNA is used: since it is perhaps common due to the coupling of these two genes (lines 98-99).
Response 3: Thank you for your comments. When a part of the HPV-DNA is integrated into the host genome, HPV E6 and E7 proteins are expressed, which is believed to be central to the transformation process. Therefore, we performed HPV E6/E7 mRNA ISH analysis in all high-risk (HR) HPV-positive samples to assess an etiological role for the development of prostate cancer, as transcription of E6 and E7 generally occurs together. These comments have been added to the introduction section.
Comments 4: Section 2.5. For the same reason, lines 109-114 should preferably be moved to the Introduction, since this is well-known literary information.
Response 4: In accordance with reviewer’s suggestion, this sentence was moved to the Introduction section.
Comments 5: Section 2.6. Line 131. The first time GG is mentioned, and it should be spelled out as Grade Group.
Response 5: The first time GG (in section 2.6) was spelled out as Grade Group.
Comments 6: Lines 90, 106 and 127 have 2+, and the tables have ++. This needs to be standardized.
Response 6: All of them were standardized as “2+”.
Comments 7: Section 3.1. Line 137. No antigen specified (preferably abbreviate as PSA).
Response 7: I revised it to “prostate-specific antigen (PSA)”.
Comments 8: Line 141. What is (<T2), abbreviate tumor? Clarification is needed.
Response 8: I revised it to “Tumor category”.
Comments 9: Why are the numbers designated as n in Table 1 and as N in Table 2?
Response 9: All of them were standardized as “n”.
Comments 10: Section 3.2. Lines 167-168. Arithmetically, 80% of 15 patients will equal 12 patients, not 9.
Response 10: This is a mistake. I revised it to “60%”.
Comments 11: In Table 1, GG is GRADE GROUP
Response 11: I revised it to “Grade Group”.
Comments 12: In table 3 – GG is Gleason group. Should be unified
Response 12: I revised it to “Grade Group”.
Comments 13: There is no reference to either tables or figures, which are not discussed at all.
Response 13: I have revised the Discussion section and included references to the tables and figures.
Reviewer 4 Report
Comments and Suggestions for Authors
The role of HPV in prostate cancer has been extensively investigated from epidemiological, pathophysiological, and etiological perspectives. Nevertheless, further studies are needed to clarify its involvement in male genital tract pathologies. In this context, the article “Human papillomavirus prevalence in the prostate and seminal vesicles: Does this virus have an etiological role in the development of prostate cancer?” focuses on two key aspects: the localization of HPV and its potential etiological role.
In the study, the authors analyzed paraffin-embedded tissue from individuals with varying grades of prostate cancer, including 157 prostate and 156 seminal vesicle samples. Methods included nested PCR (nPCR) for HPV detection, flow-through hybridization for genotyping, in situ hybridization (ISH) for viral localization, and in situ analysis of HPV E6/E7 mRNA expression with p16 immunohistochemistry in HR-HPV–positive tissues.
Certain limitations are noted, such as the genotyping kit not covering all HR-HPV types and the gene expression analyses being restricted to HR-HPV–positive tissues. Nevertheless, these do not diminish the study’s value, which provides important insights. Some points requiring further clarification are highlighted in the review comments attached to the revised manuscript.
Kind regards

Comments on the Quality of English Language
The English could be improved to more clearly express the research.
The comments are in the revised manuscript
Author Response
Thank you for your kind comments and suggestions. I revised the manuscript according to your comments, and all of corrections were described directly in the revised manuscript. Replies for your comments were included in PDF file.

Round 2
Reviewer 1 Report
Comments and Suggestions for Authors
The authors revised the manuscript correctly following my comments. This paper can be published in Viruses
Author Response
Comments; The authors revised the manuscript correctly following my comments. This paper can be published in Viruses
Reply: Thank you for reviewing our manuscript. I am very happy to be able to receive your kingly suggestions and comments, and then quality of our paper was likely to be improved. Thank you again for your assistances.
Reviewer 2 Report
Comments and Suggestions for Authors
The authors have tried to address the concerns I had communicated with them in my first review. However, without some kind of a normal negative control for side by side comparison, the data to support the conclusions is still absent in the revised manuscript. I had suggested in my comments earlier that cultured normal human prostate cells (not tissues as stated by the authors) could technically be used as a negative control for repeating the experiments. I am afraid that without proper negative controls this manuscript is not ready to published.
Author Response
Comments 1: The authors have tried to address the concerns I had communicated with them in my first review. However, without some kind of a normal negative control for side by side comparison, the data to support the conclusions is still absent in the revised manuscript. I had suggested in my comments earlier that cultured normal human prostate cells (not tissues as stated by the authors) could technically be used as a negative control for repeating the experiments. I am afraid that without proper negative controls this manuscript is not ready to published.
Response 1: Thank you for your comments. We really agree with your suggestions. While this approach may be considered in theory, currently no standardized method exists to infect these cells with HPV. Furthermore, HPV-induced carcinogenesis generally requires chronic infection over many years, and cultured cells cannot fully reflect the prevalence or distribution of HPV genotypes in normal prostate tissue. For these reasons, we think that such cells may not provide an adequate control group for the present study soon. This represents a clear limitation, and we have added the following sentences to the Limitation section.
Line326-332: In addition, we collected all of samples from the patients with prostate cancer in the present study, and a normal control group was not included. This limitation makes it difficult to establish a baseline prevalence in healthy prostate tissue, and leads to difficulty in distinguishing between the mere presence of the virus and its oncogenic etiological role. Therefore, further studies including non-cancerous tissues are required to solve this limitation. Alternatively, uses of cultured normal human prostate infected with HPV cells as a control may be considered.